# Three-Dimensional Printing Using Recycled High-Density Polyethylene: Technological Challenges and Future Directions for Construction

**Faham Tahmasebinia** [1,2,*] **, Marjo Niemelä** [1] **, Sanee Mohammad Ebrahimzadeh Sepasgozar** [3] **, Tin Yiu Lai** [2] **, Winson Su** [2] **, Kakarla Raghava Reddy** [2] **, Sara Shirowzhan** [1] **, Samad Sepasgozar** [1] **and Fernando Alonso Marroquin** [2]

[1] Faculty of Built Environment, The University of New South Wales, Sydney, NSW 2052, Australia; marjo.niemela@unsw.edu.au (M.N.); s.shirowzhan@unsw.edu.au (S.S.); sepas@unsw.edu.au (S.S.)

[2] School of Civil Engineering, The University of Sydney, Sydney, NSW 2006, Australia; tlai4178@uni.sydney.edu.au (T.Y.L.); wisu6606@uni.sydney.edu.au (W.S.); reddy.chem@gmail.com (K.R.R.); fernando.alonso@sydney.edu.au (F.A.M.)

[3] Division of Software Engineering, Danesh Bonian Institute, Khorasan Shomali 0915, Iran; sanee.mes@gmail.com

* Correspondence: F.tahmasebinia@unsw.edu.au; Tel.: +61-(02)-9351-2171

**Abstract:** Three-dimensional (3D) printing technologies are transforming the design and manufacture of components and products across many disciplines, but their application in the construction industry is still limited. Material deposition processes can achieve infinite geometries. They have advanced from rapid prototyping and model-scale markets to applications in the fabrication of functional products, large objects, and the construction of full-scale buildings. Many international projects have been realised in recent years, and the construction industry is beginning to make use of such dynamic technologies. Advantages of integrating 3D printing with house construction are significant. They include the capacity for mass customisation of designs and parameters to meet functional and aesthetic purposes, the reduction in construction waste from highly precise placement of materials, and the use of recycled waste products in layer deposition materials. With the ultimate goal of improving construction efficiency and decreasing building costs, the researchers applied Strand 7 Finite Element Analysis software to a numerical model designed for 3D printing a cement mix that incorporates the recycled waste product high-density polyethylene (HDPE). The result: construction of an arched, truss-like roof was found to be structurally feasible in the absence of steel reinforcements, and lab-sized prototypes were manufactured according to the numerical model with 3D printing technology. 3D printing technologies can now be customised to building construction. This paper discusses the applications, advantages, limitations, and future directions of this innovative and viable solution to affordable housing construction.

**Keywords:** 3D printing (3DP); construction processes; architectural design; concrete engineering; numerical modelling; arch roof; high-density polyethylene (HDPE); additive manufacturing (AM); computer-aided design (CAD); manufacture; design; sustainability

---

## 1. Introduction

3D printing technologies, known broadly as additive manufacturing (AM) processes, fabricate three-dimensional structures from CAD files by adding successive layers of materials. The emergence of advanced digital technologies has led building construction practitioners to rethink the cost- and resource-effectiveness of their productivity, and make decisions towards adopting new

technologies [1–3]. Many industries—from manufacturing to medicine—have made use of 3D printing technologies since the 1980s. However, architectural practices and the construction industry have only recently begun to adopt these technologies for construction [4].

3D techniques can be described in terms of four main manufacturing processes: (i) subtractive manufacturing; (ii) additive method (rapid prototyping); (iii) forming techniques; and (iv) hybrid methods. Subtractive manufacturing refers to cutting technologies that remove undesirable parts of objects using CNC (Computer Numerical Control) machines or similar tools. AM refers to the addition of materials to an object or the creation of a new object from design in the absence of human intervention. Forming techniques involve the reshaping of objects without reducing or adding materials, and hybrid manufacturing processes involve a combination of the best features of both subtractive and additive techniques [5].

3D printing, which is the focus of this paper, was first associated with a specific AM process. AM is now widely recognised as a disruptive technology that could transform the construction industry, with 3D printing technologies and materials reconstructing traditional manufacturing methods, whether in a revolutionary or evolutionary way.

3D printing, as a technique for constructing matter, has become popular in recent years. Using a virtual and digitally constructed 3D design model as its blueprint, it sequentially lays down layers of material until the envisaged physical object has been created. The technique is cost effective and saves time. When compared with traditional concrete construction, it saves on labour costs and offers a higher curing rate, due to the nature of the materials used. Various additive manufacturing techniques (e.g., extrusion, jetting, sheet lamination, and photopolymerisation) are used to fabricate light-weight and large-scale composite products that combine polymers and fillers (e.g., carbon fibre, glass, CNTs (Carbon nanotubes), etc.) [6–8]. These are highly efficient for manufacturing, and minimise both costs and waste. Thermoplastics used in the process include polyethylene, polypropylene, polycarbonate, polyvinylchloride and acrylonitrile butadiene styrene (ABS).

There exist a variety of automated additive manufacturing methods but, among them, 3D inkjet printing has proved to be highly efficient. It draws from different computer-design morphologies (e.g., 2D or 3D), offers a continuous extrusion process, and has the capacity to cure areas designed in polymer/filler suspension. 3D printing has been used to build such functional components as large buildings, car products, medical devices (human tissues, organs, and dental material) and wearable gut. It also has applications in other branches of production, and the manufacturing of acoustic and vibration products.

The potential advantages of AM in housing are significant. They include not only improved efficiencies pertaining to the environment and financial resources but, also, the capacity to customise designs for aesthetic and structural applications to increase architectural freedom. Previous studies suggested drivers for automation, and digital technology adoption includes an improvement in productivity and safety, and a reduction in construction costs and time [9–11]. Many recently completed projects—such as Contour Crafting in the United States, Apis Cor in Russia, and Winsun in China—have provided evidence that the 3D printing of houses can be realised on various scales, and AM technologies offer innovative solutions for affordable housing construction.

While various scholars have described the benefits of AM, "rapid building prototyping" (RBP) practices have not been sufficiently explored from a construction perspective. The construction industry is complicated by the variety of building components created: their sizes, quality, and safety, and their strength as used in different projects. Scholars must, therefore, extend their RBP practices to different scales in the laboratory, and on actual construction sites. This study applies a structural analysis perspective to the design of a roof prototype that uses particular materials.

Recent studies have tended to develop and apply frameworks to test the performance of fresh printing mixtures in the laboratory [12]. Such studies have tended to create small prototypes of building designs to understand potential challenges and deficiencies associated with RBP. This project

aims to analyse the structural feasibility of a 3D-printed building prototype by using the finite element software Strand 7. It also investigates the feasibility of building a 3D-printed house prototype.

Previous studies have generally looked at 3D printing in different contexts, and presented the technology's advantages. However, its practical limitations—for example, its inability to produce steel bars and other complicated building elements—have not been fully investigated. Overall, its advantages include layer-by-layer deposition, which negates the need for formwork, and moulds that lead to a reduction in construction waste. Highly accurate placement means that materials can be considered as more efficient materials for construction. However, previous studies have failed to fully introduce and investigate replacements for steel, particularly from a practical perspective. In situ 3D printing has the potential to cut transport costs, energy consumption, and pollutant emissions, as well as injuries and fatalities [13]. However, such potential benefits should be evaluated in the contexts where they are critical to practitioners. Furthermore, while construction companies use a variety of material mixes—including cement, steel, and fibres—and 3D printing has the capacity to introduce multi-material deposition, including different percentages of plastics and fine materials, there is insufficient evidence to show that all construction materials can be mixed and used by 3D printers. This is of significant importance, since variations in the material composition at different stages of a print could reduce material use if it is not structurally required.

This paper aims to demonstrate that the combination of highly efficient materials with recycled waste products, in the formation of structural housing elements, is desirable, saving construction companies costs while protecting the environment by introducing waste solutions. In addition, the paper uses laboratory 3D printing to examine a smaller scale of its proposed roof elements for the feasibility of their shape stability. Although the process is simple, it is efficient enough to provide solutions for remote areas, where supplies and skilled labour can be hard to come by.

For a range of high-end applications, such as 3D-printed buildings and products used in the aerospace and automotive industries, it is essential to understand the mechanical properties of materials [14,15]. Mechanical properties pertain to a material's ability to withstand applied loads and displacements. Underlying these properties is a constitutive law that relates the strain experienced by the material to an applied stress. This strain is measured using tensile tests: a sample is loaded with tension, and the strains are measured as a function of applied stress to determine such properties as stiffness, strength, ductility, and toughness. Stiffness relates to the elastic modulus and defines the force required to produce elastic deformation. Three factors are critical to the design of novel, high-strength materials: chemical composition, nano/microstructure, and architecture.

Various materials have been used in 3D printing for a range of applications. Kazemian et al. [12] used a mixture of cementitious materials and nano-clay, or polypropylene fibre or silica fume, to study the workability of such 3D-printed mixtures on a construction-scale project in terms of print quality, shape stability, and printability window. Another concrete printing machine consisted of a 5.4 m × 4.4 m × 5.4 m frame, and a printed head on a movable horizontal-beam that shifted in the Y and Z directions while the printing head moved only in the X direction [9,16], and time-dependent structural build-up of cement materials in a layer-by-layer construction process [17]. A fibre/cement mixture (cement content: 827 kg/$m^3$) was used to print a 2 m concrete bench with the compressive strength of 110 MPa, and the shear strength of 0.3–0.9 kPa [18]. Rushing et al. [19] tested the physical properties of components developed from 3D printing by applying extruder testing, drop table tests, compressive strength, and flexural strength. Furthermore, various mixed concrete materials (e.g., fillers, wood fibres, polymers, and geopolymers) have been used to print 2D and 3D components for building applications [20–25]. Nakagawa et al. [26] used the technique to fabricate 3D products using carbon fibre (tensile strength: 5.3 GPa) combined with ABS (tensile strength: 30 MPa) and without cement. However, limited work has been carried out on the printing of 3D building materials that use polymer/nanocarbon-based composites.

In December 2016, one 3D house printing manufacturer [27] successfully utilised mobile 3D printing technology to create the first house printed solely from 3D printed materials. It was built on

site in Moscow, with a pure machine printing time of 24 h and a cost of $10,000, as stated by Apis Cor [27]. The area of the printed build was only 38 m$^2$—it aimed to serve as a capability showcase—but the time it took, and the total cost, were extremely advantageous when compared with traditional methods. Since such 3D-printed buildings arise from the implementation of an innovative technology to construct a physical model, making use of a relatively new method rather than traditional practices, they are not yet common place.

At the time of writing this report, there were no strict guidelines or related Australian standards for the design of 3D-printed buildings. As a consequence, the functions and limitations of 3D printing in construction are closely linked to a contractor's knowledge. Existing Australian standards for structural strength, however, provide a reliable guide to the design of structural integrity in a modelled structure. In particular, AS1170 and AS3600 can be used as design guidelines.

This paper begins by reviewing the literature and categorising advances in and applications of AM. Second, it presents a specific roof prototype and discusses various materials before analysing the designed prototype with the structural analysis software Strand 7. Third, the paper uses virtual information and converts it into a physical object that represents the roof. Finally, the paper discusses the lessons learnt and the challenges to this mixed design-built practice on a laboratory scale. It also suggests future directions.

## 2. Three Printing Advances and Applications

Today, there exist a variety of 3D layer deposition processes. These include fused deposition modelling (FDM), which heats a plastic filament, such as PLA (Polylactic acid) or ABS (Acrylonitrile Butadiene Styrene), to a viscous state that bonds, layer by layer, to create a geometry; selective laser sintering (SLS), which binds powder into a solid by applying layers of binder and powder, in sequence, to build a form; and stereolithography apparatus (SLA), or digital light processing (DLP), which use a vat of photoreactive liquid resin to form thin layers that stack up to create a shape. SLA uses ultraviolet (UV) light to photochemically set the resin, while DLP projects a digital image. With an ever-increasing selection of technologies and materials available, 3DP has expanded beyond the initial idea of generating design iterations and models [28]. However, the construction industry still faces a mammoth challenge in scaling up existing 3D printing technologies [29].

3D printing in concrete is rapidly developing—from proof of concept by Contour Crafting in the United States in 2001, to the printing of a five-storey apartment complex by Chinese company Winsun in 2015 [4], and Apis Cor in Russia, now promoting its mobile system for 3D-printed houses. Construction-scale 3D printing falls into two broad areas: 3D printing of components that are assembled onsite, and in situ 3D printing that uses a programmable Cartesian coordinate system to deposit material layers onsite. Both systems have advantages and limitations—the scale and size of building elements can limit the use of either to a singular 3D printing strategy. For this project, we propose a hybrid of the two systems, namely, in situ 3D printing of roof components that are then assembled on site.

A detailed survey of available additive deposition systems is beyond the scope of this paper, but a brief overview, here, focuses on the advantages and limitations of technologies that have previously been utilised in construction to build homes.

### 2.1. In Situ System

The first concrete extrusion gantry system, combined with a material process solution, was developed and patented by Khoshnevis, from the University of South California in the United States, under the name Contour Crafting, in 2001 [30]. Contour Crafting uses an on-site 3-axis gantry system, the same system that is used by a small-scale 3D printer, to deposit layers of a cement-based paste against trowels that create a smooth surface. The use of two trowels gives rise to an accurate and smooth finish, with a key feature being the ability to build utility conduits into walls [31]. Doors and windows are fitted after the printing construction has been completed. In situ gantry systems have various limitations in large-scale projects. These include their expense and complexities related to the

transport and installation of a heavy gantry system, as well as the size of the gantry's build envelope or reach of the truss extension for depositing material.

## 2.2. Contour Crafting on a Gantry

### 2.2.1. Mobile Rotating Manipulator with Extension Arm

Apis Cor was the first company to develop a mobile 3D printer for the construction industry that comprised a rotating truss on a central base. It claims to have successfully printed a 37 m$^2$ house on site in 24 h [13]. The Apis Cor has a maximum operation area of 132 m, with an 8.5 m reach from the central manipulator and a maximum height of 3100 m. The printer uses the same cement mixture for printing foundations and walls, thereby reducing material waste.

### 2.2.2. Offsite Gantry System

Winsun in China uses a factory-based gantry system to create 3D-printed building components that are then transported by truck on site. The printer extrudes a concrete mix, layer by layer, through a nozzle and wall. Floor and ceiling components are created with a diagonal truss pattern, leaving hollow sections intended for insulation [24]. As formwork is not required, almost endless design iterations are possible, limited only by the size of the gantry and the infrastructure. Offsite or factory-based gantry systems do not need to be transported, eliminating the expense of installing heavy infrastructure and freeing the size of the build envelope from restrictions that arise from transportation needs. However, its transportation needs and challenges are akin to those of traditional prefabricated houses, including the movement of large components.

### 2.2.3. In Situ Cable Suspended Platform

Since the objects produced range in size from building components to full-scale houses, a system has been developed under the generic label "cable-suspended platform" [29]. A cable-suspended platform consists of an end-effector that is manipulated by automated motors via multiple cables that are attached to a rigid external frame. One advantage of a cable-suspended platform is that it provides a larger work envelope while being relatively inexpensive to transport. The system is comparatively lightweight, and can be disassembled and reassembled on site, offering further flexibility for larger builds.

## 3. Research Scope and Method

This report includes a design prototype for possible structural layouts of a 3D-printed house, as shown in Figure 1. The layouts resemble existing houses, with a living room, bedrooms, garages, etc. Dimensions were chosen to provide residents with enough area for their daily activities. Numerical results from FEA (Finite Element Analysis) software Strand 7 have been included to demonstrate the structural strength of the design, and these have been checked against Australian standards for residential housing. Structural analysis was also performed to investigate the advantages of an arch roof over a typical flat-slab roof for 3D-printed houses.

### 3.1. 3D-Printed House Prototype

The envisaged location of the 3D-printed house for load analysis was generic, and the load analysis was designed as an ultimate limited state, with the worst-case scenario formulated according to what a residential house would experience in Australia. The structural design resembles an arch, spanning a width of 22.3 m, a height of 6 m, and a depth of 17 m. An arch-like structure was chosen because it reduces tensile stress. This design decision was critical because steel bar reinforcement could not be incorporated into the 3D-printed structure. This made it essential to reduce the tensile stress experienced by the concrete structure, since concrete has a low tensile strength. Figure 2 shows each structural member within the house. It presents the truss-like internal layouts of the roof slab. The

arch consists of warren-truss-like members to provide greater interior structural strength than that of a simple arch. This layout also reduced the self-weight of the roof, minimising the loadings that it needed to withstand.

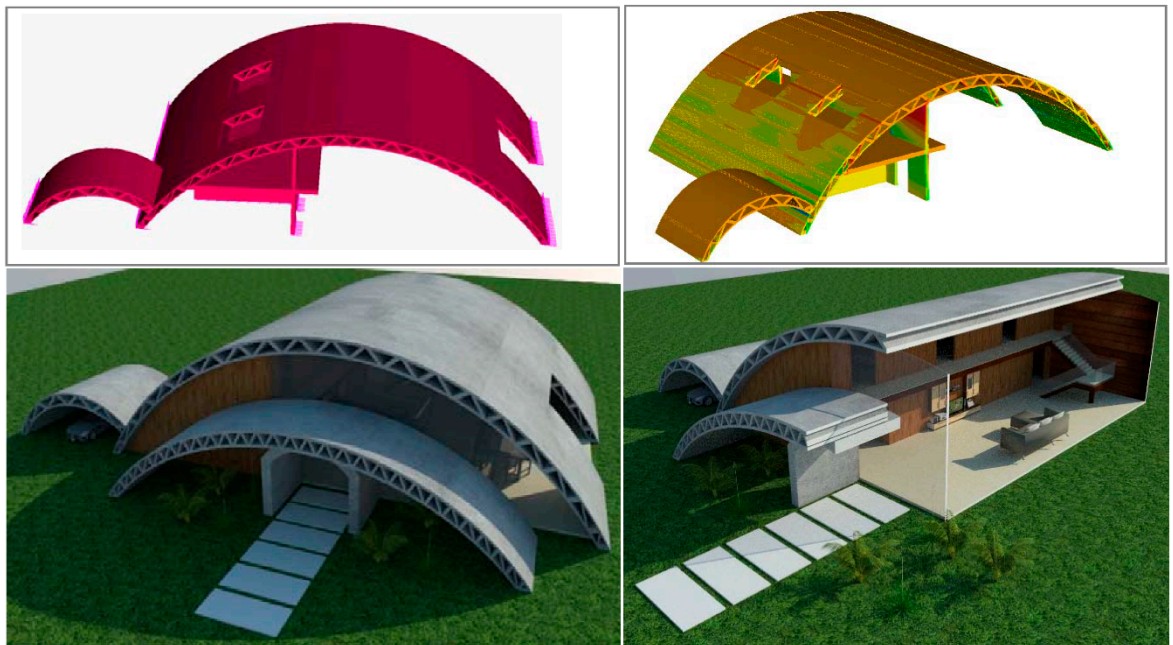

**Figure 1.** Graphical abstract of 3D-printed buildings.

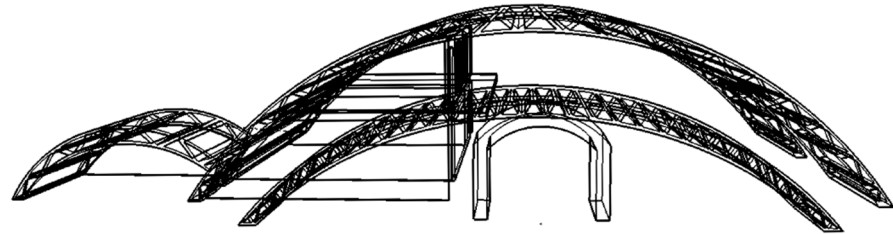

**Figure 2.** Sketch of the conceptual design for the wire frame of the 3D-printed house model, showing its internal truss-like structure.

All members were printed using a 3D house printer. The ground floor and first floor consist of a printed floor slab. An arch-shaped roof spans the entire structure. Supporting the arch is a truss-like member, which further strengthens the roof. There is also a shear wall that extends from the ground floor to the first floor and up to the main roof. The arch on the left-hand side is for use as a garage, and shares a similar shape to the main roof. The small arch at the front acts as an architectural façade only, and is not considered to be part of the main structural member. Figures 3–5 show the front elevation, side elevation, and plan view of the 3D-printed house. All dimensions are in metres.

The 3D-printed house in Figure 1 was designed after considering all the daily activity areas that can be found in a traditional house, from living and dining areas to the master bedroom with bathroom ensuite, and even a storage room on the ground floor. The small roof on the left-hand side, as mentioned above, acts as a garage. A staircase connects the ground floor to the first floor where two bedrooms and a bathroom can be found, as shown in Figure 5. Three small windows (opening size: 1 m × 1 m) in the main arch roof of the first floor serve as skylights for the upstairs rooms. There is also an opening of 4.5 m × 3 m in the main arch roof (as shown in Figures 3 and 4) that serves as an entrance to the garden envisaged for the right-hand side of the house, as shown in Figure 3. These design decisions were made before any numerical analysis, so that the numerical model assessed a practical house design that was fit for daily use. All of the openings stated above were included in the

Strand 7 numerical modelling, so that their effects on the strength and stability of the structure could be investigated.

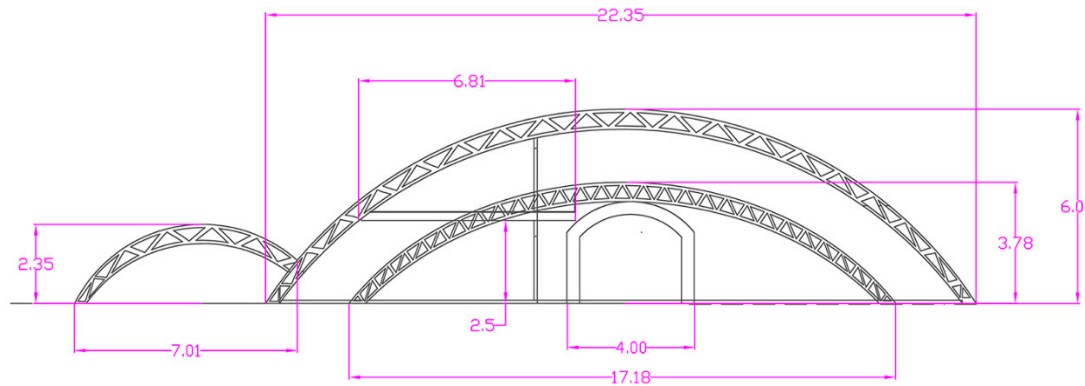

**Figure 3.** Front elevation of the 3D-printed house.

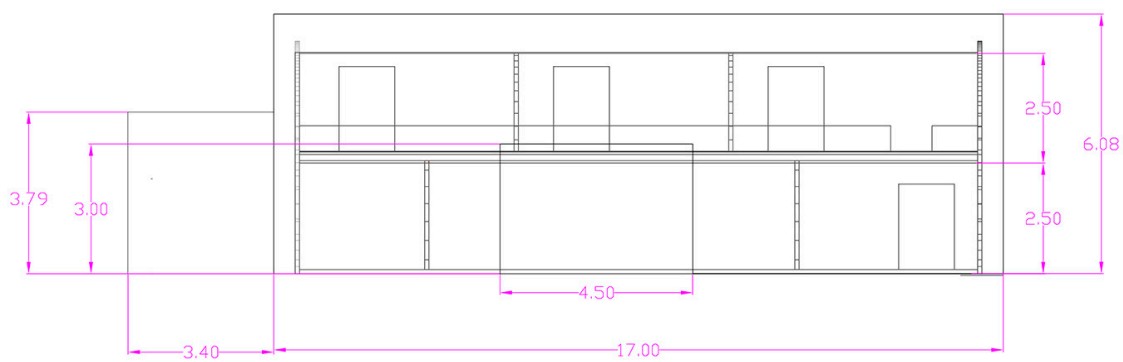

**Figure 4.** Side elevation of the 3D-printed house.

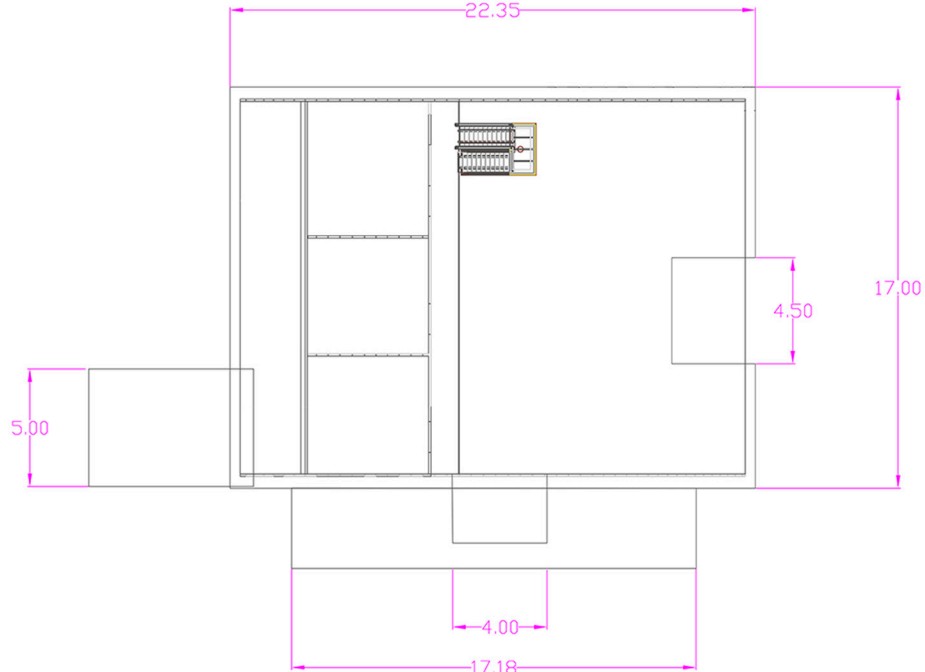

**Figure 5.** Plan view of the 3D-printed house.

## 3.2. Material Choice

The main material for the structural component of the building was concrete. Since large aggregate particles cannot pass through the nozzle of the 3D printer, the material size of the aggregates was limited to a diameter of less than 20 mm. Another limiting factor of 3D printing in concrete construction is the inability to use steel bar reinforcement; because cement materials are poured out from the 3D printer nozzle, it is unfeasible to incorporate reinforcement within the structure. Therefore, the tensile strength of concrete becomes a limiting factor in designing a 3D-printed house. To add strength in this instance, the cement paste mix was combined with recycled high-density polyethylene (rHDPE). According to Dym and Williams [32], rHDPE increases both the tensile strength and the serviceability of the concrete. Table 1 summarises the differences between plain concrete (C1) and concrete that is mixed with rHDPE fibres (C2–C4). It shows how the addition of these fibres results in a higher overall tensile strength than plain concrete. This is beneficial since the cement mixture in 3D-printed houses lacks other reinforcements.

**Table 1.** Concrete strengths [32].

| Concrete Property | Plain | Ø 0.25 mm Fibres | | |
|---|---|---|---|---|
| | C1 | C2 | C3 | C4 |
| **Elastic Modulus** | | | | |
| $E_c$ | 24.2 | 24.5 | 24.9 | 25.2 |
| **Compressive Strengths** | | | | |
| $f_{ck}(cube)$ | 33.2 | 34.3 | 31.1 | 32.3 |
| $f_{ck}(cube)$ | 38.1 | 40.1 | 38.4 | 37.7 |
| $f_{ck}(cyl)$ | 23.3 | 26.2 | 24.1 | 23.4 |
| **Tensile Strengths** | | | | |
| $f_{ct}(cyl)$ | 2.79 | 3.08 | 2.95 | 2.96 |
| $f_{ct}(cyl)$ | 3.32 | 3.47 | 3.49 | 3.43 |
| $f_{ctm}$ | 3.84 | 4.35 | 4.14 | 4.37 |

The difference between C2 and C4 reflects the difference in the mixture ratio of fibre and air, as summarised in Table 2. The percentage of fibres is denoted in the third column of Table 2, where it represents the overall percentage of rHDPE fibres in the cement mixture. For example, the use of 0.4% in C2 can achieve an excellent balance between the highest tensile strength and the compressive strength, with a slightly less elastic modulus than the other mixture ratios. Different mixture ratios alter the compressive and tensile strengths of concrete for different applications. For this 3D-printed house, C2, as shown in Tables 1 and 2, they will be chosen to maximise the tensile strength of the cement mix, which is the desired property for the limiting factor.

**Table 2.** Concrete mix properties [32].

| Concrete Series | | Fibres | Slump | Air |
|---|---|---|---|---|
| | Label | % | (mm) | % |
| **Plain Concrete** | | | | |
| | C1 | - | 65 | 3.2 |
| **FRD (Fibres: Ø$_1$ = 0.25 mm, L$_1$ = 23 mm)** | | | | |
| | C2 | 0.4 | 36 | 3.4 |
| **(r$_{a1}$ = 92)** | C3 | 0.75 | 22 | 3.3 |
| | C4 | 1.25 | 17 | 3.4 |

The addition of HDPE fibres to concrete mix can also affect its durability. There are number of the microfibers in rHDPE to be added to concrete mix for 3D printing. These microfibres are small enough to be compatible with the 3D house printer nozzle. By adding them, the overall concrete elastic

modulus becomes 25.2 GPa. Its density, on the other hand, decreases to 2380 kg/m$^3$. The nominated rHDPE diameter is 0.25 mm, and the strengths are indicated in Table 2. The use of recycled HDPE has the added benefit of ameliorating the environmental impact of construction materials because it is commonly downcycled from other plastic wastes. The resultant cement mix also offers significant environmental benefits over steel mesh because it is recyclable, whereas reinforced concrete is usually hard to recycle.

### 3.3. 3D House Printer

Apis Cor [27] developed a 3D house printer that was suitable for this project. It can print structural layers of concrete mix for walls, slabs, beams, and roofing, and is very flexible in terms of design modelling. Due to current technological limitations, it is not yet possible to print an entire house on one setting; multiple layers of materials must be set up over multiple runs. The size and length that the 3D printer could reach at the time of writing was limited by its physical size, hence the need for multiple settings to create a residential house of typical size. However, it was believed that there would soon be a more compatible 3D printer that was large enough to manufacture house models more efficiently.

### 3.4. 3D Construction Sequences and Techniques

The laying sequences of the cement mix are shown in Figure 6. The printer would manoeuvre around the site, depositing different layers as shown in Figure 6a. Each horizontal section would be printed layer by layer. For example, the base of the arch roof and shear walls would be printed from base to top until there were layers that required additional support, due to their position being suspended in the air; for example, the first-floor slab of the 3D-printed house design. Multiple scissor lifts would be placed under the area that required additional formwork support, and a horizontal formwork would be suspended by the scissor lifts to increase the working height of the printer. The scaffolding, as shown in Figure 6b, would be erected to allow the printer to layer the floor slab. This process would be repeated until the entire structure was printed.

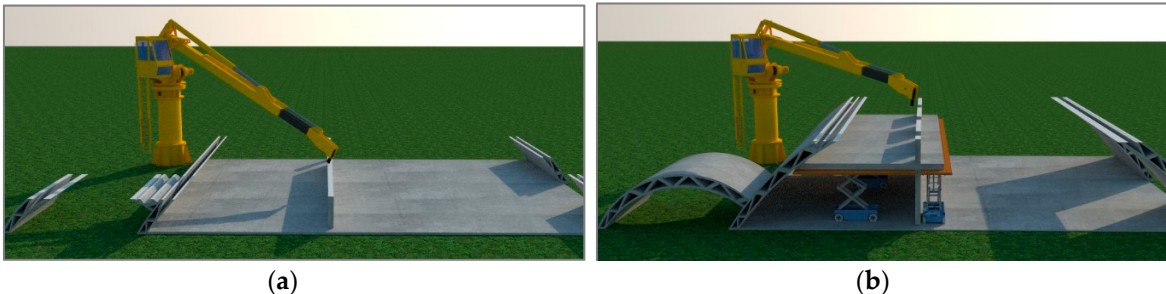

|     |     |
| :-: | :-: |
| (**a**) | (**b**) |

**Figure 6.** Proposed construction sequence for suspended layers.

This technique is called fused disposition modelling (FDM). It requires that the printer combine different raw materials together, and generates a paste mixture to form the structure. At present, this method of printing houses is the one that has been used by Apis Cor.

Another technique that is common in 3D printing, particularly on a smaller scale, is stereolithography (SLA). SLA exploits ultraviolet laser light by shining a laser beam on to a liquid resin—a bath of chemicals—to create a 3D model. The model is then cured in a different chemical bath, and a reductant support part that was created during the printing process is removed. This method has a fast printing rate: it can print the entire structure on one setting. However, there are problems pertaining to material limitation and curing. No research on 3D modelling traces the use of concrete mixtures with this printing method. Also, the space needed to produce and cure the structure might not be justifiable economically. However, the method can manufacture models with high precision,

and the printer does not need multiple settings, which is a limiting factor for the printer developed by Apis Cor [27].

## 4. Results and Discussion

### 4.1. Numerical Modelling

Finite element software, Strand 7, was used to conduct a numerical analysis of the design structure that is the subject of this paper. The overall structure consists of multiple arch roofs that contain triangular voids. These form a truss-like structure inside the arch roofs, taking advantage of the transfer of loads within a truss structure and reducing self-weight. Smaller sized triangular voids at the end of the structure provide greater strength while reducing the possibility of buckling.

The front elevation of the model was first constructed in AutoCAD, due to the structure's complexity. The roof consists of multiple triangular voids within a curved arch. This is impossible to model by hand with regular square elements in Strand 7, so the structure was first modelled as a 2D vector model in AutoCAD. The 2D model was constructed by following the dimensions in Figure 7. The resulting model was imported as a dxf file, and into Strand 7 as a face element. This face element is shown in Figure 8. It shows the main arch roof, the side arch roof, the garage, the shear walls within the main roof, and the first-floor slab.

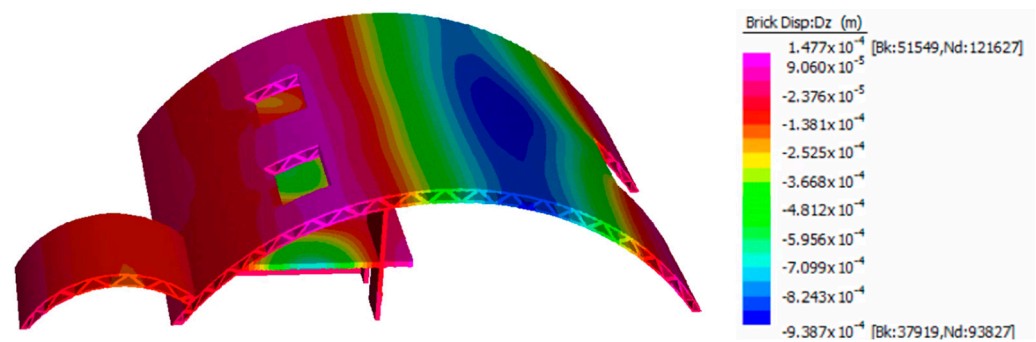

**Figure 7.** Displacement of the roof due to gravity load (self-weight).

Next, the auto-meshing function of Strand 7 was adopted to create the best mesh for the model, as computed by Strand 7. The mesh sizes were controlled to generate mesh within a dimension of less than 0.15 m so as to increase the accuracy and converge the numerical result. Quad 4 elements and Tri 3 elements were used in generating the front face shell elements. Then, the shell elements were extruded to brick elements, with the same length increments as the dimensions of the Quad 4 elements. After obtaining the 3D model that was formed by brick elements, further modification was made within Strand 7, including the design of door and window openings, constraints for the base supports, defining material properties of the concrete–rHDPE mixtures, defining loads and load combinations, etc.

### 4.2. Load Combinations

The combination of actions used to design this 3D-printed house would satisfy the load combinations outlined in AS1170.0 [33]. The house was designed to the ultimate limit states that are used for checking stability, with the following combination factors given in AS1170.0 [33] Clause 4.2.1. Note that, as this structure is assumed to be located in the Sydney area of Australia, the significance of earthquake load is minimal. According to AS1170.4 [33] in domestic housing structures with heights under 8.5 m, or an importance level of 1, the significance of earthquake load is outweighed by other loads. As a result, any earthquake load imposed on the structure is not included in this analysis. In the following load combinations, the symbol G represents permanent action, Q represents imposed

action, and $W_u$ represent wind action. These are the load combinations used in Strand 7 to check the structural design action as stated in the Australian Standards:

1.35G
1.2G + 1.5Q
1.2G + $W_u$ + 0.4Q (positive wind)
1.2G + $W_u$ + 0.4Q (negative wind)
0.9G + $W_u$ (positive wind)
0.9G + $W_u$ (negative wind)

### 4.3. Gravity Load

Self-weight is defined as the weight of the structure itself, and was calculated by including gravity in the load combination. The density of the brick elements had to be defined, as well as the concrete–rHDPE mixtures. The density of the mixture is slightly less dense—at 2380 kg/m$^3$—due to the addition of rHDPE fibres, than typical concrete mix, which has a density of 2400 kg/m$^3$. In Strand 7, 9.81 m/s$^2$ is taken as the acceleration due to gravity in the Z direction. Figure 7 is a graphical representation of the displacement scale of the structure purely due to gravity load.

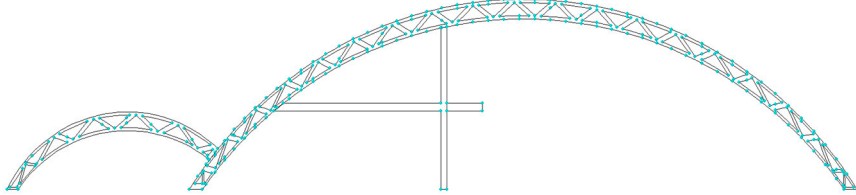

**Figure 8.** Face element model created in Strand 7.

The overall loading effect, in this case, is owing to self-weight, which includes gravity and is vertically downwards. The contour shows that maximum deflection occurs in the main roof, near its right-hand side. This is due to the large opening and lack of support from shear walls. The large opening in the right-hand side of the roof, which forms an entrance to the garden from the house, may also contribute to this deflection. However, the deflection is at a minimum, at only 0.94 mm, and falls within the acceptable range according to AS3600 [34]. Although there are window openings on the left-hand side of the roof, this area does not seem to experience concentrated stress because it is well supported by the shear walls, in accordance with the design decision. This dead load is used at a later stage as a loading combination in accordance with AS1170.0 [33].

### 4.4. Imposed Action

Imposed action is the sufficient allowance for vertical impacts that arise from the usual movement of people and shifting of furniture. In accordance with AS1170.1 [33], the designed location is defined as Group A1 (domestic and residential activities, self-contained dwellings). The imposed action for Group A1 should satisfy the follow requirements.

For the ground floor, the imposed action shall be designed as a uniformly distributed action of 1.5 kPa, or concentrated action of 1.8 kN if the area is greater than 320 mm$^2$, for the calculation of punching and crushing shear. Since the house has a ground floor of 374 mm$^2$, both the uniformly distributed action and the punching/crushing shear will be considered in the calculations.

For the first floor, the imposed action shall be designed as a uniformly distributed action of 1.5 kPa, or a concentrated action of 1.8 kN if the area is greater than 320 mm$^2$, for the calculation of punching and crushing shear. Since the first floor has an area of 102 mm$^2$, calculation of punching/crushing shear is not required.

For the stair area, the imposed action shall be designed as a uniformly distributed action of 2.0 kPa, or concentrated action of 2.7 kN.

For the non-habitable roof space, the imposed action shall be designed in accordance with AS1170.1 [33]. Since the roof is defined as R2—a roof that uses structural elements—the uniformly distributed actions shall be calculated as:

However, the uniformly distributed load shall be no less than 0.25 kPa. Since the area projected by the roof is 379.46 m$^2$:

$$q = \frac{1.8}{A} + 0.12 \tag{1}$$

$$q = \frac{1.8}{379.46} + 0.12 \tag{2}$$

$$q = 0.1247 \le 0.25 \tag{3}$$

Therefore, the imposed action shall be taken as 0.25 kPa. It shall also satisfy a concentrated load of 1.4 kN for the requirements of punching/crushing shear. Figure 9 is a graphical representation of the stress contour of the structure after all imposed action had been incorporated into Strand 7.

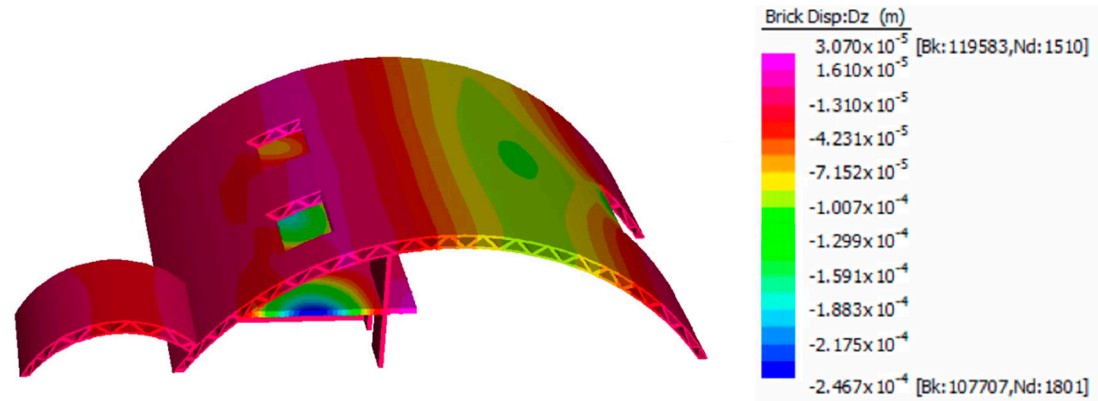

**Figure 9.** Displacement due to imposed action.

The live load areas are defined as concentrated only in habitable areas (i.e., the floors and first-floor slab), with minimal loadings in inhabitable areas (i.e., the arch roofs). Therefore, the deflection contour due to imposed action shows a different profile when compared with the deflection due to dead load in Figure 9. The maximum deflection occurs within the first-floor slab, with a value of 0.25 mm. The differing nature of both load cases is due to the way AS1170.1 [33] defines imposed action according to an allowance of the effects of vertical impact from live load. The deflection by imposed action is minimal, and within the acceptable range. The difference in load concentration area is advantageous to prevent loadings concentrated at a single spot during load combinations with dead loads. This load case is used at a later stage for the load combinations.

*4.5. Wind Load*

The 3D-printed house was assumed to be located in the Hornsby area of Sydney for the sake of load analysis. Therefore, in accordance with AS1170.2 [33], it would be classified as Region A2. It was assumed that the topography was flat, and there was no shielding effect. The building was in terrain category 3. The dynamic response factor for such a building was conservatively taken as 1.0. For the designed build, there exists a large door on the west side with dimensions of 4.5 m wide and 3.06 m high. There are also three windows on the east side, with dimensions of 2 m by 2 m.

The building was designed to face a wind event with an average recurrence interval of 500 years. It was designed for wind from a northerly direction. Area reduction and combination factors were ignored, and taken as 1.0 for the ease of calculation, and as part of a conservative approach.

First, the design wind speed needed to be established from the given data. The design parameter of the roof height is 6.08 m. For Region A2, the wind average recurrence $V_{500}$ = 45 m/s. For wind coming from the north, the critical $M_d$ is 0.95 from a northwesterly direction. The terrain multiplier $M_{z,cat}$ was taken as 0.83 for a roof at 6.08 m within terrain 3. Therefore, the designed wind speed was taken as

The wind speed at full roof height should therefore be taken as 39.76 m/s. For the external wind pressure, it should be calculated in accordance with AS1170.2:2002, where for an arch roof with r/d = 3/22 = 0.14:

$$V_{des} = V_R \times M_d \times (M_{z,cat} \times M_s \times M_t) \tag{4}$$

$$V_{des} = 45 \times 0.95 \times 0.93 \times 1 \times 1 \tag{5}$$

$$V_{des} = 39.76 \tag{6}$$

$$C_{p,e} = -(0.55 + 0.2 \times \frac{h}{r}) \tag{7}$$

$$C_{p,e} = -0.578 \tag{8}$$

Therefore, the external pressure $p$ is found by

$$p = 0.5 \times \rho_{air} \times V_{des}^2 \times C_{fig} \times C_{dyn} \tag{9}$$

where $C_{fig} = C_{p,e}k_ak_ck_ek_p$, and $k_a, k_c, k_e, k_p$ shall be taken as 1.0 for the most conservative case. Therefore,

$$p = 0.5 \times 1.2 \times (39.76^2) \times (-0.578) \times 1 \tag{10}$$

$$p = -0.548kPa \tag{11}$$

For the leeward wall, for d/b = 17/22 = 0.77, $C_{p,e}$ is taken as −0.5. Therefore, leeward wall pressure is calculated as

$$p = 0.5 \times 1.2 \times (39.76^2) \times (-0.5) \times 1 \tag{12}$$

$$p = -0.474kPa \tag{13}$$

For side wall pressure, in accordance with Table 3, pressure differs according to the location of the wall, with consideration of the maximum roof height. Table 3 summarises the wall location and its corresponding wind pressure.

**Table 3.** Side wall pressure and its corresponding wall region.

| Wall Region with Respect to Roof Height | Wall Region with Respect to Actual Distance Away from Windward Wall | Side Wall Pressure (kPa) |
|---|---|---|
| 0–1 h | 0–6.08 m | 0.616 |
| 1–2 h | 6.08–12.16 m | 0.474 |
| 2–3 h | 12.16–17 m | 0.285 |
| >3 h | N/A (as depth of structure is 17 m) | 0.190 |

Figure 10 shows the graphical representation of the displacement scale of the structure due to wind load. The deflection contour due to wind load shows a different profile when compared with the other load case. The main difference is that wind loads are applied perpendicular to the surface of the roof, instead of being a vertical load. Wind loads are only applied on the roof, and not on the shear walls or first-floor slab. Therefore, the loading effects are higher in areas that are exposed to the outer environment which, in this case, are the arch roofs. Maximum deflection occurs at the main roof slab, with a value of 0.11 mm. This is the least out of three of the load cases, which is intuitive since wind has a minimal effect on low-rise buildings. Wind loads will often be a concern only in high-rise buildings. This load case is used at a later stage for the load combinations.

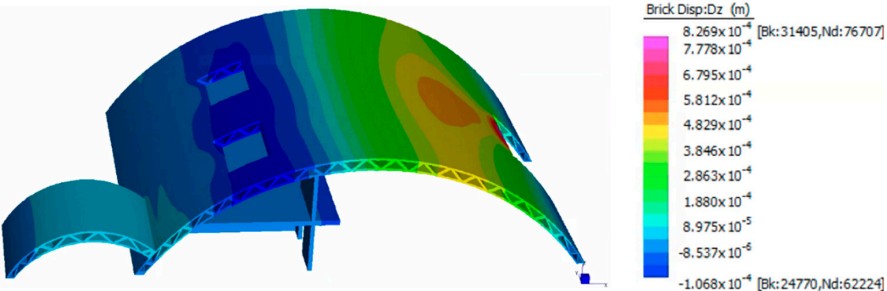

**Figure 10.** Displacement due to the applied wind load.

## 4.6. Load Combination Result

All six different loading combinations in accordance with AS1170.0 [33], with different multiplier factors, were fed into Strand 7 and compared. Figure 11 shows the resulting stress bar for the maximum tensile and compressive stresses that the structure experienced among all load combinations. Both are from a load combination of 1.2G + 1.5Q, where Figure 11a shows the stress in yy and Figure 11b shows the stress in xx directions. Both figures also include the contour for inspecting the load concentration point among the building's design. As stated previously, the minimal effect of wind loadings arises because low-rise buildings expose a limited surface area, and there is minimal wind speed at a low height. The resulting critical load combination case is, therefore, one without wind loadings, and its influence is minimal.

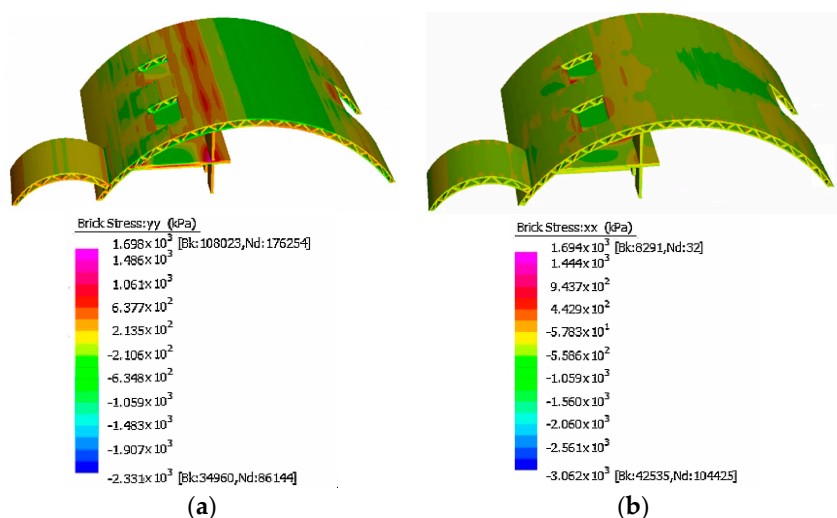

**Figure 11.** Maximum bending stress among all load combinations in two xx and yy directions.

Note that the maximum tensile stresses in both xx and yy directions are very similar, and differ by less than 0.01 MPa. The maximum tensile stress in yy direction is 1.698 MPa and, in xx direction, is 1.694 MPa. Those two values are less than the tensile characteristics of the concrete–rHDPE mix used for the structure, which exceed 2.8 MPa in tensile strength. The maximum compressive stress among all load combinations, on the other hand, is 3.063 MPa, which is significantly smaller than the compressive characteristic of concrete mix, at 26 MPa. Therefore, the structure would withstand the load combinations that a typical house would experience, according to AS1170 [33]. As shown in Figure 11a, the concentration of stress is mostly along the first-floor slab and the roof, which are sufficiently supported by the shear walls near the middle. If not for the shear wall in the middle, which allows for the large roof span, the numerical result might have been different, and exceed its tensile strength due to an unsupported span. The use of rHDPE mix is, again, highly advantageous because it increases the tensile strength of the concrete and enables it to withstand most of the tensile stress without the addition of reinforcements.

### 4.7. Structural Design

It was in our best interests to investigate the benefits of adapting an arch design for this 3D-printed house. A limiting factor of this project was the inability to use a reinforcement bar within the cement mix. Therefore, arch design comprised the main design concept because of its ability to reduce tensile stress. Hand calculations of the main arch roof were performed, as well as an analytical estimation of arch-like structures [32]. The calculations assessed the behaviour of the arch under gravitational line loads, with the use of the small angle theorem among other assumptions. Dym et al. [32] discussed the underlying assumption while performing this analytical solution. The main analytical method used for this report is noted, with Figure 12 showing the parameters required. An arch is defined by radius $R$ and semi vertex angle $\alpha$ by design. The parameter q represents the force per unit length, $z$ represents the roof height, $v$ and $w$ the force vector along the arch, and theta $\theta$ the angle along the arch.

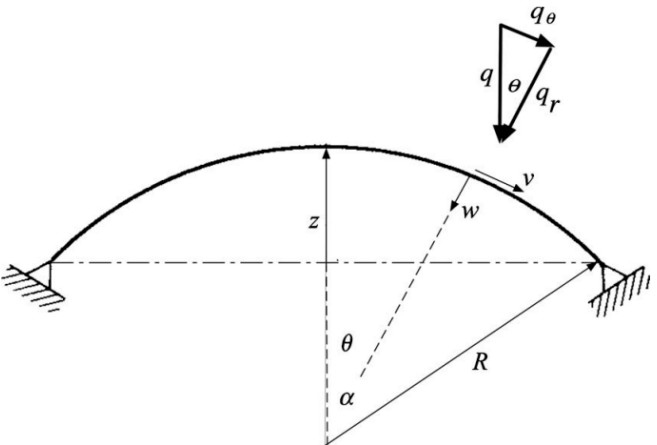

**Figure 12.** Geometry of and loading on a thin circular.

Figure 13 shows the design parameters for the roof arch of the 3D-printed house. The truss-like voids within this arch were not considered in the hand calculations (the arch is assumed to be fully filled as a solid concrete roof). Therefore, the gravity load for the roof would be overestimated in this hand-calculation, and over-conservative. It should also be noted that the depth of the roof is 17 m, as per the design in the 3D model within Strand 7 (Figure 13).

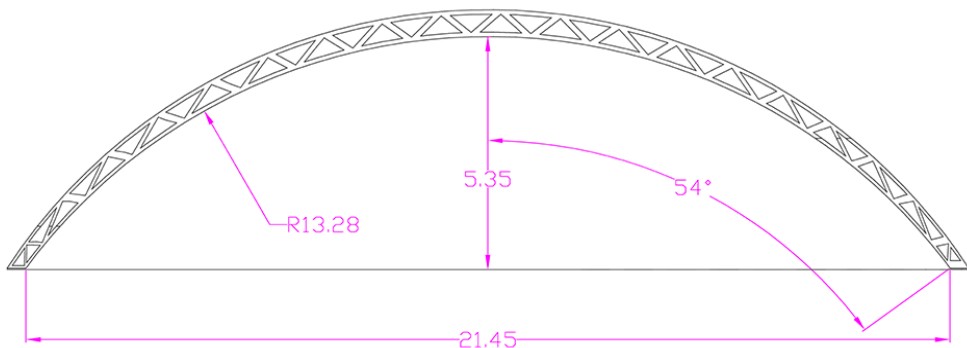

**Figure 13.** Geometry of the designed 3D-printed house.

From the analytical formula by Dym et al. [32], the arch rise parameter λ is given by the ratio of the roof rise as

$$\lambda = \frac{f}{h/2} = \frac{2R}{h} \times (1 - \cos\alpha) \tag{14}$$

and the moment resultant along the roof arch is given by

$$M^c(\theta) = \frac{q \times l^2/24}{1 + 4\lambda^2/15} \times \left(1 - 3 \times \left(\frac{\theta}{\alpha}\right)^2\right) \tag{15}$$

By substituting the arch rise parameter $\lambda$, the bending moment can be obtained. Figure 14 shows a plot of the bending moment stress along roof angle $\theta$. The roof is mostly subjected to compressive stress along the two roof-ends, with the middle subjected to tensile stress. The maximum tensile stress is about 1.6 MPa, which the concrete roof can withstand because the 3D-printed house uses a concrete–rHDPE mixture that has a tensile characteristic of 3.0 MPa. Therefore, the concrete required no reinforcement. This result is consistent with the numerical analysis from Strand 7, where the maximum tensile stress experienced by the arch roof was found to be 1.69 MPa. Note that the referring result arose from a load combination instead of a pure dead load by the roof. However, since this hand calculation assumed a solid roof-slab, rather than a roof full of truss-like voids, the result of the hand calculation should be taken as an overestimate of the bending stress experience by the roof itself (see Figure 14).

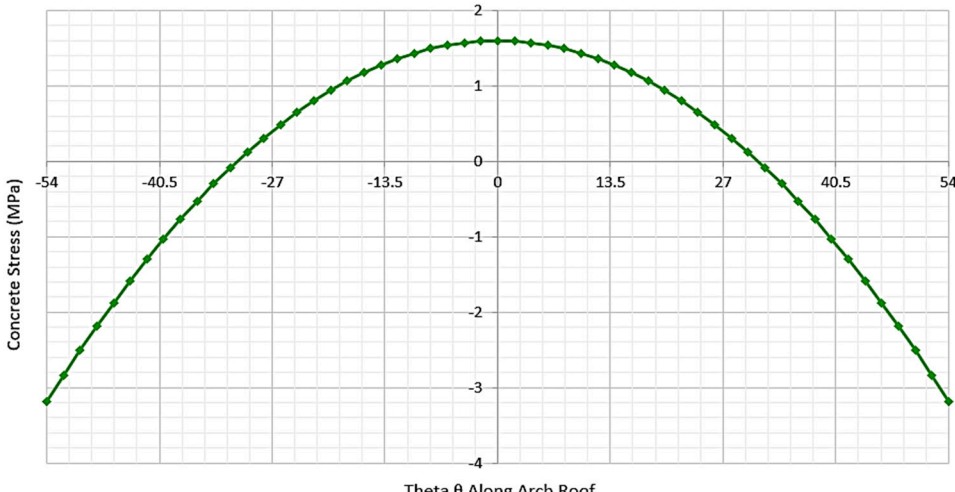

**Figure 14.** Bending moment along arch roof $\theta$.

This result was compared with a traditional flat-slab roof for the purposes of investigating and comparing the benefits of using an arch roof in this design. A typical flat-slab roof is used for this calculation to find the maximum bending moment within the roof slab. Figure 15 shows the typical setup of a flat-slab roof with the same dimensions (width and height) as the arch roof. The width of the roof and column are the same as the arch roof.

The calculations for finding the bending moment within the flat-slab roof are performed using the simply supported beam theory, with each end of the slab assumed to be pin-jointed. The formula for the stress along the slab, at distance $x$, is given by the analytical solution of

$$M(x) = \frac{q \times x}{2} \times (L - x) \tag{16}$$

The plot of bending stress along the slab is shown in Figure 16. The tensile stress is higher than that of the arch roof. The maximum bending stress that the flat-slab roof can experience is 10 MPa at the middle of the roof. This result was expected since the flat slab would have maximum sagging in the middle, particularly in the absence of any support. The long span of the roof induced an excessive tensile stress that was much greater than the tensile strength of the concrete–rHDPE mixture, which is only 3 MPa. Such high tensile stress in the middle of the flat slab is typically acceptable in traditional reinforced concrete. However, for this 3D-printed house, the excessive tensile stress does not suit

the cement mix in the absence of reinforcement. Therefore, a flat-slab roof design is not suitable for 3D-printed housing. The excessive tensile stress would cause the concrete to fail.

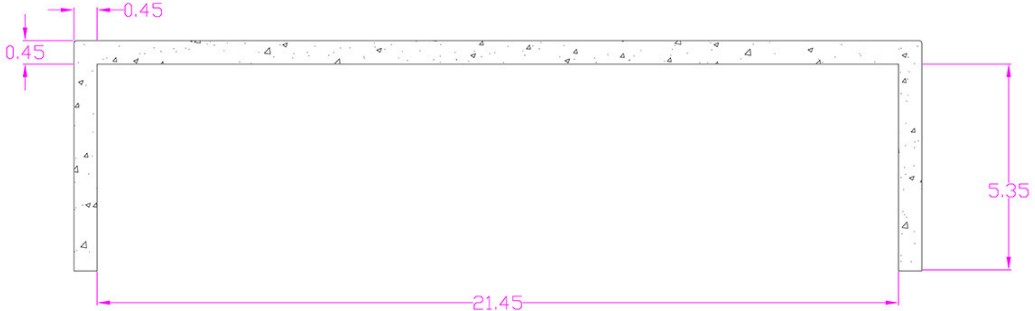

**Figure 15.** Geometry of a typical flat-slab roof.

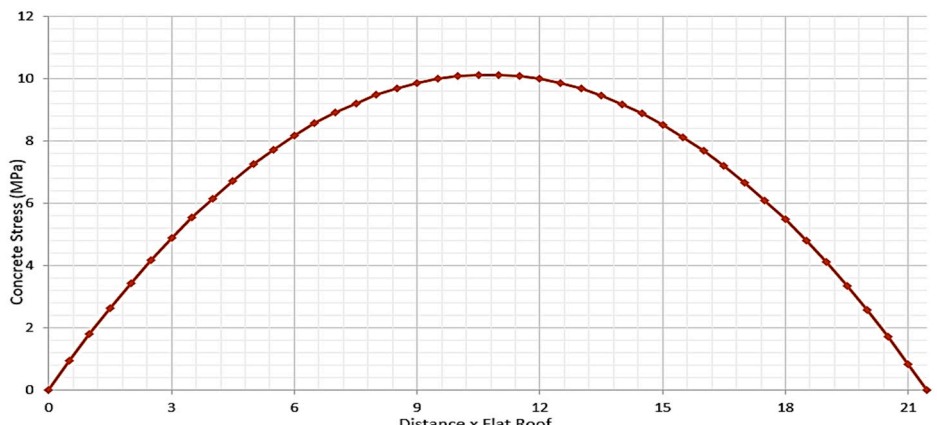

**Figure 16.** The bending moment along the flat-slab roof.

Special attention was devoted to preparing and assembling the numerical models developed using 3D printing. Initial preparations, which included drafting the relevant executive sketches of the roof as well as other members, were undertaken, with precision, in AutoCAD. The prepared AutoCAD files were considered AS input files for the 3D printing machine.

*4.8. Structural Design*

Figure 17 illustrates different stages in the creation of the 3D model. As presented, different elements of the roof were added, step by step, to the initial elements, to complete the drafted curved roof. Further details can be seen in Figure 18. One of the main contributions of the current paper is to display the capacity of 3D printing techniques to create structural details, including diagonal and bracing members that are available in the developed numerical models.

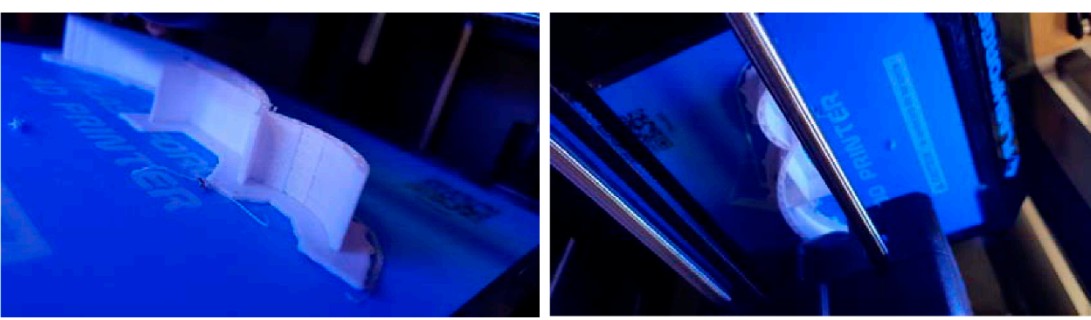

**Figure 17.** Different stages in the creation of the 3D model.

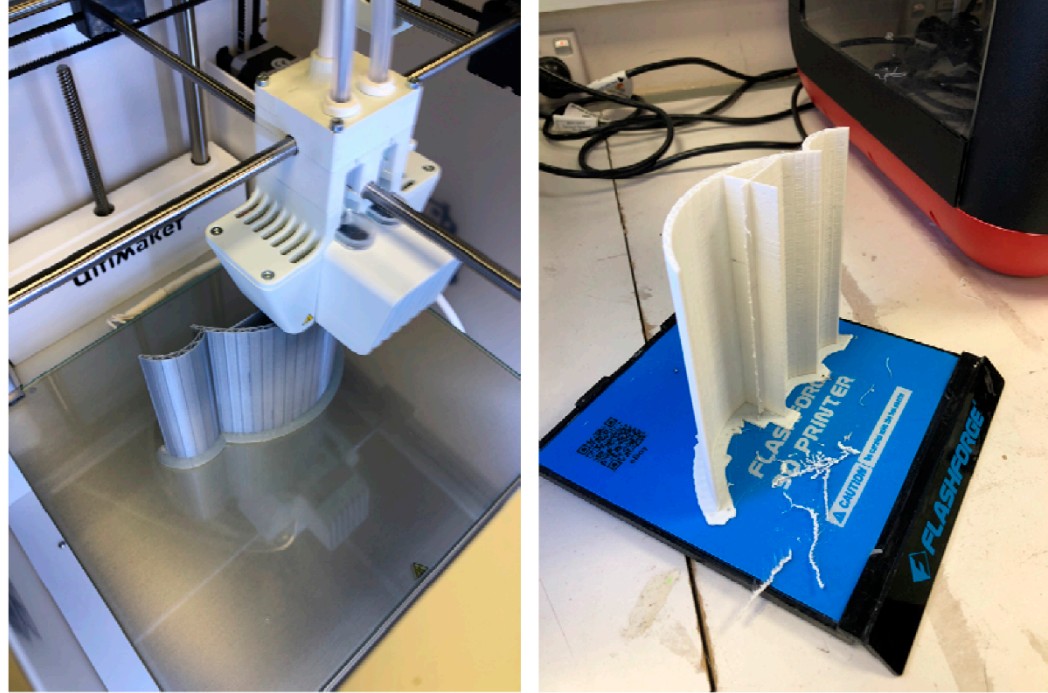

**Figure 18.** A representation of the process of creating the model in the lab, including the main arch roof.

## 5. Discussion

This study contributes to the design of a numerical model that uses a cement mix material combined with the recycled waste product HDPE. It differs from previous studies in two ways. First, the presented design procedure and analysis of an arched truss-like roof has been carefully analysed to evaluate the feasibility of constructing this case, complicated by the need for steel reinforcements and specific shape requirements. This paper shows that the limitations of 3D printing, through which it is not easy to produce steel bars in situ, can be addressed by making use of HDPE materials. Second, the paper analysed the shape stability of this design, and a small-sized model was developed to evaluate its feasibility.

With the application of Strand 7 Finite Element Analysis, it was found that the construction efficiency could be increased, and the overall building costs would fall as a result of implementing the designed material and 3D printing. Strand 7 software was used because it is commercially common in the Australian region, and previous 3D printing studies had ignored both its use and Australian standards in their analytical methods. This had raised uncertainties about the use of 3D printing in practice in this geographic region.

The specific roof design presented in this paper for a 3D-printed building was also used to manufacture a prototype for further laboratory testing of shape stability. Shape stability was a major criterion suggested by a recent publication for the laboratory testing of fresh printing mixtures [12]. Initial experiments have shown that 3D printing technology is highly customisable for various sites, terrains, and to meet the aesthetic requirements of design, even though the efficiency of the method, including the production time and logistics, should be investigated in future studies. Using this technology, a formwork system is not required for layer-by-layer deposition. The size of a building or modular item is, therefore, not restricted, and any material dimensions and an almost infinite array of geometric shapes are achievable. Furthermore, individualised design that does not incur the costs of traditional construction variations can lead to cost reductions and higher customer satisfaction. However, the benefits of AM in construction have largely been studied from the viewpoint of the end-user, even though the benefits of these technologies and processes present an opportunity to rethink decisions made in the design process [29].

Previous studies have discussed other applications for 3D printing, from material perspectives and size limitations [35]. However, detailed investigations of the whole process—from virtual data and the reproduction of a physical object to becoming a reality—are scarce. This study aimed to use both Strand 7 and CAD information to develop a prototype that was complicated in terms of its proposed materials and shape. Additionally, Labonnote [29] investigated potential improvements in material science, both in terms of construction and improving material properties, for additive manufacturing.

This study differed from previous works by carefully investigating the whole process of converting virtual data into a physical model at different scales, and reporting the results of the laboratory experience. It is crucial that results be investigated further to provide a more rigorous framework for understanding the virtual data and its real objects. This study also argued that we still need to extend our theoretical understanding of the process of shifting from the virtual to reality by testing all new, advanced software and hardware technologies, as practitioners try to use larger machines to produce larger objects. Such a possibility is still in doubt, however, since the real size does not necessarily equate to large, because modular construction tends to use smaller parts of items and assemble them in real construction sites. Xu [36] recently discussed the results of two printed half individual plinths, and demonstrated that it is possible to create small objects that use convenient AM machines for building maintenance.

The present study faced several limitations. The proposed design considered the real size roof, and the analysis focused on a numerical modelling of the roof, applying Strand 7 Finite Element Analysis software. The design aimed to serve as an investigation into the feasibility of building such a roof, in a way that would satisfy Australian standards for 3D printing, with a cement mix that incorporated the recycled waste product HDPE. In this section, the real size of the structure was considered. However, the second part of the paper tended to create a prototype of the roof for evaluating its form and shape stability in the context of AM techniques. In this case, the limitation was the size of the printer. While the size of the machine limited the evaluation of the prototype, it was, though, helpful in understanding how different design methods and materials affect the final product. The main contribution of our experiments was identifying the challenges of prototyping that can assist practitioners to know what to avoid when they intend to create a real-sized roof. In fact, the central contribution of this study is the evaluation of RBP practices, which refers to emerging technologies for the use of design data (e.g., CAD and Strand 7) in fabricating building objects quickly and economically. Previous studies have tried to evaluate rapid prototyping techniques in a different context, but many shapes and construction objects have been ignored [35]. We used three different printing machines, two of them being smaller sized 3D printers that produced a roof model with a maximum 9.5 cm span, and the third being a robot that produced a 30 cm span. Printing limitations pertaining to size have been reported in previous studies [37] but, recently, scholars had suggested continuous laboratory experiments to learn more about the possibilities of producing different building parts [12].

One of the limitations of 3DP construction is the number of materials available for structural AM [31]. In terms of concrete building structures, there are commonly two main ways to build: one involves pouring concrete on site; the other involves precast concrete slabs/beams being delivered to the site. For pouring concrete on site, curing under typical site conditions, can only achieve about 65% of the concrete's compressive strength in seven days, attaining in excess of 90% of its compressive strength after 28 days [38]. Precast concrete requires intensive transport and labour costs, and offers limited design flexibility because the concrete is cast in a factory. Regardless of which method is used, intensive labour and time are required. With the increasing cost of labour and time needed to finish construction work by hand, conventional house construction methods would become only more expensive over time. Therefore, it is a major challenge to produce a stable, printable material with properties such as consistent or predictable setting times, stability during construction, and effective bonding between layers. Particularly for in situ manufacturing, material behaviours must be thoroughly investigated under a range of conditions to achieve a robust product that will not only withstand the structural load upon completion but, also, during the fabrication process.

## 6. Conclusions

This paper reviewed and assessed the viability of 3D printing technology in house construction. It demonstrated how a model could be created and prepared for a site practice, and showed and discussed the prototype design and lab experiments. This construction technique is a recent innovation; there is no extensive industrial experience or past practice to report. Researchers, such as Ninoslav [21], began to experiment with the idea of concrete polyethylene mix in 3D printing techniques for concrete house construction, but studies have shown that a concrete mixture with rHDPE enhances the tensile strength of the concrete, and that this is viable as a 3D-printed material. As well as being environmentally friendly and reducing the carbon footprint of recycled polyethylene, the increased overall tensile strength of the mixture compensates for the lack of tensile reinforcement that arises from using cement paste in 3D printing. Existing research had suggested such materials would be crucial to choosing materials suitable for commercial 3D-printed housing.

Australian Standards were used, here, to check the structural response under load combination, and to investigate the feasibility of 3D house printing technology in construction. As no specific standard pertains to 3D construction techniques at present, this report followed the AS1170 [33] series and AS3600 [34] to guide design decisions. A main challenge for 3D-printed houses need to overcome is the lack of steel bar reinforcements within the structure. Reinforcements are common in modern structures, but are not feasible in 3D printing. Therefore, tensile stress becomes a limiting factor. The structure considered in this paper incorporated the recycled substance rHDPE as a building material. It was found that increases in tensile strength are crucial in helping the concrete structure resist load combinations. The numerical result showed that the overall highest tensile stress the structure experienced was 1.698 MPa, which was manageable as a result of the rHDPE mix. The most critical loading case was found to be the self-weight of the structure, followed by the live loads. These were expected, due to the nature of the house: a low-rise building with minimal activities that would contribute to live loads. Most of the stresses, therefore, arose from withstanding the structure itself. Stresses induced by wind loadings were calculated to be small due to the minimal wind speed at a low height. Therefore, the hollowed truss-like arch roof helps to reduce the stresses that would otherwise contribute to the dead loads.

The key design concept here was the use of an arch roof as the main spanning slab for the entire 3D-printed house. Such arch-like structures have long been known to reduce tensile stress over long spans [32]. The arch shape allows roof loadings to be transferred with a curved load path down to the supporting edges and, hence, effectively transfers most of the loadings as a compressive stress. The structural response of the arch-roof building was found to fall within an acceptable range. As shown by hand calculations, the use of a flat-slab roof would not be feasible because the maximum tensile stress that the concrete can withstand is beyond the acceptable limits of the concrete tensile strength. However, the physical property of arch elements successfully reduces the maximum tensile stress for the concrete, so it can withstand the tensile stress without the aid of reinforcements. The Strand 7 results show, clearly, that the structure would sustain different load combinations without failing. Gravity load alone causes the maximum deflection on the structure. The maximum compressive and tensile stresses resulting from different load combinations would also not exceed the ultimate strength. This suggests that the material choice and structural orientation of the model were appropriate. There are several ways to construct house models with 3D printing at present. However, all of them have limiting factors that render the use of 3D-printing technology in house construction impractical. Further investigation and the development of 3D printing technology, in future, is bound to provide society with a better and faster method of building houses.

**Author Contributions:** Conceptualization: All authors; Literature review: K.R.R., M.N., S.S. (Sara Shirowzhan), S.M.E.S. Research Method: All authors; Software: T.Y.L., W.S., K.R.R., S.M.E.S., F.T., F.A.M. Drafting the article based on the experimentation reports: F.T., S.S. (Samad Sepasgozar), M.N., S.S. (Sara Shirowzhan). Resources and interpretation: F.T., M.N., S.S. (Samad Sepasgozar), F.A.M.

**Funding:** This research received no external funding.

**Acknowledgments:** Authors would like to express deepest appreciation to the University of New South Wales and the University of Sydney to provide the convenient places to undertake the current research.

**Conflicts of Interest:** There is no conflicts of Interest in this research.

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
