# Peer review of "Three-Dimensional Printing Using Recycled High-Density Polyethylene: Technological Challenges and Future Directions for Construction"

_buildings, doi:10.3390/buildings8110165_

Round 1

Reviewer 1 Report

This manuscript summarized the broad applications of the AM in the construction field and described detailed information regarding how the virtual model transforms to physical object. The authors also briefly discussed challenges and future directions.

It appears that recycled HDPE in the concrete increases tensile strength. Regarding the use of recycled HDPE, does it alter any innate physicochemical properties when recycled?

Are there any other materials that may offer comparable properties?

As the title of this article includes the sustainability aspect, can the authors comment on sustainability and economic advances using this materials more in details?

Please revisit a list of references to unify its listing style.

Some legends for the sectioned figures are needed to be modified (e.g. (a) and (b) are needed to be separately explained).

Author Response

Reviewer’s comments ( Reviewer 1)

Authors’ explanations

This manuscript summarized the broad applications   of the AM in the construction field and described detailed information   regarding how the virtual model transforms to physical object. The authors also   briefly discussed challenges and future directions.

1)      It   appears that recycled HDPE in the concrete increases tensile strength.   Regarding the use of recycled HDPE, does it alter any innate physicochemical   properties when recycled?

2)        Are there any other materials that may offer   comparable properties?

3)        As the title of this article includes the   sustainability aspect, can the authors comment on sustainability and economic   advances using this materials more in details?

4)        Please revisit a list of references to unify its   listing style.

5)      Some   legends for the sectioned figures are needed to be modified (e.g. (a) and (b)   are needed to be separately explained).

1)      Yes, this is correct.  Recycled   HDPE can lead to changing both physical and mechanical properties as well as   chemical condition in the materials.

2)      Any kind of brittle materials can be used which   would be comparable.

Examples of brittle materials include cast iron, concrete, and some glass products.

3)        This topic came up with an idea about the   optimization. In this research both thermal and mechanical optimization in   lab scale sample numerically checked. The main advantages is to suggest a   technique where would be fast and efficient in construction and the input   materials economically would be justified. The suggested technique can   present future construction in the countryside and rural areas.

4)        All of the available references were revised.

5)        Further explanations were individually added.  

Reviewer 2 Report

This paper dealt with a methodology for 3D printing using recycled high-density polyethylene in construction processing and architectural design. In addition, the authors showed their technological challenges and sustainability. The proposed method is well explained and the case study also illustrates the validity of the research.

Author Response

Reviewer’s comments ( Reviewer 2)

Authors’ explanations

This paper dealt with a methodology for 3D   printing using recycled high-density polyethylene in construction processing   and architectural design. In addition, the authors showed their technological   challenges and sustainability. The proposed method is well explained and the   case study also illustrates the validity of the research.

The   authors would like to deeply express their appreciations.

Reviewer 3 Report

The paper presents a research on three-dimensional printing by using cement mix incorporated with recycled waste product High Density Polyethylene (HDPE). The methodology applied is interesting, the organization is good and the text is sufficiently clear.

Author Response

Reviewer’s comments ( Reviewer 3)

Authors’ explanations

The paper presents a research   on three-dimensional printing by using cement mix incorporated with recycled   waste product High Density Polyethylene (HDPE). The methodology applied is   interesting, the organization is good and the text is sufficiently clear.

The   authors would like to deeply express their appreciations.

Reviewer 4 Report

Minor language check required.

Author Response

Reviewer’s comments ( Reviewer 4)

Authors’ explanations

Minor language check required.

A   comprehensive (language) proof reading was conducted.
